# Variant-specific antibody correlates of protection against SARS-CoV-2 Omicron symptomatic and overall infections

José Victor Zambrana [1,2,8], Ian A. Mellis [3,4,5,8], Abigail Shotwell[1], Hannah E. Maier [1], Yara Saborio[2,6], Carlos Barillas[2], Roger Lopez[2,6], Gerald Vasquez[2], Miguel Plazaola[2], Nery Sanchez[2], Sergio Ojeda[2], Isabel Gilbertson[1], Guillermina Kuan[2,7], Qian Wang[3], Lihong Liu[3], Angel Balmaseda[2,6], David D. Ho [3] ✉ & Aubree Gordon [1] ✉

Vaccination and prior infection elicit neutralizing antibodies targeting SARS-CoV-2, yet the quantitative relationship between serum antibodies and infection risk against viral variants remains uncertain, particularly in underrepresented regions. We investigated the protective correlation of pre-exposure serum neutralizing antibody levels, employing a panel of SARS-CoV-2 pseudoviruses (Omicron BA.1, Omicron BA.2, and ancestral D614G), and Spike-binding antibody levels, with symptomatic BA.1 or BA.2 SARS-CoV-2 infections and overall infection, in 345 household contacts from a SARS-CoV-2 household cohort study in Nicaragua. A four-fold increase in homotypic-neutralizing (e.g., BA.1-neutralizing vs. BA.1 exposure) titers was correlated with protection from symptomatic infections (BA.1 protection: 28% [95%CI 12–42%]; BA.2 protection: 43% [20–62%]), and ancestral-neutralizing titers were also correlated with protection from either variant, but only at higher average levels than homotypic. Mediation analyses revealed that homotypic and D614G-neutralizing antibodies mediated protection from infection and symptomatic infection both from prior infection and vaccination. These findings underscore the importance of monitoring variant-specific antibody responses and highlight that antibodies targeting circulating strains may be more predictive of protection from infection. Nevertheless, ancestral-strain-neutralizing antibodies remain relevant as a correlate of protection. Our study emphasizes the need for continued efforts to assess antibody correlates of protection.

Some people exposed to SARS-CoV-2 become infected shortly thereafter, while others do not. Humoral immunity, elicited by prior infection or vaccination, is important for protection from infection[1]. At a population level, several groups have shown that serum anti-SARS-CoV-2 antibody titers are an informative correlate of protection from infection in clinical trials and observational cohorts[2–9]. However, the extent to which serum anti-SARS-CoV-2 antibody titers correlate with protection is highly variable, explaining 48.5% to 94.2% of infection risk across different vaccine efficacy studies[10]. Additionally, SARS-CoV-2 continues to evolve and increasingly evade existing antibody responses[11–13]. The extent to which such evolution compromises the interpretation of anti-SARS-CoV-2 antibody assays focused on

historical, non-predominant variants remains underexplored[14]. We hypothesize that variant-specific neutralizing antibody titers may be better correlated with protection from infections or symptoms than the standard ancestral-strain-neutralizing titers used in most assays. Also of importance to global public health, the majority of published studies of antibody correlates of protection focus on participants from North America, Europe, and Asia; relatively few include participants from Central and South America or Africa where a majority of first SARS-CoV-2 exposures were through infection, not vaccination, and less-studied vaccines were deployed[15]. Differences in early SARS-CoV-2 exposures among individuals in Central and South America, compared to other regions raise concerns that correlates of protection identified in other populations may not be applicable in the context of Central and South America.

In most studies of correlates of protection from infection, specific participant virus exposures remain unidentified, and individuals will vary widely in their exposures over time, unless prevalence is very high. Herein, we overcome this tracking problem with a household-based cohort study with an embedded transmission study, intensely monitoring "index cases" who present with infection and their household member "contacts," building on our prior study, in which we identified only clinical correlates of protection from SARS-CoV-2 infection in households in Nicaragua[16]. The present study aims to measure protection against SARS-CoV-2 BA.1 and BA.2 infection and moderate/severe infection in the Household Influenza Cohort Study (HICS) population following transmission of SARS-CoV-2 (prior infections with ancestral or Delta strains) and vaccinations. We combine household-level infection tracking data with hundreds of participants' neutralizing antibody titers against ancestral SARS-CoV-2 (D614G) and the contemporaneous exposure viral variants Omicron BA.1 and BA.2, spike-binding titers, and clinical information, including infection and vaccination histories and age, to better understand clinical and serum antibody measurements that are predictive of protection from infection or symptomatic infection risk after a given virus exposure. Finally, we integrate infection and vaccination histories with neutralizing titers in causal mediation analysis to show that neutralizing antibodies mediate protective effects of prior viral exposures from infection or vaccination.

## Results

### Variant-specific neutralizing antibody titers are correlates of protection from symptomatic and any SARS-CoV-2 infection

We measured SARS-CoV-2 antibody levels in 345 individuals from the HICS study, representing all enrolled household contacts of each index case exposed to BA.1 or BA.2 introductions, before exposure to these variants. We used these measurements to assess the protective effect of pre-existing antibodies against infection and moderate/severe infection. Households included in the study experienced the introduction of BA.1 or BA.2 SARS-CoV-2 strains by a household member, with these strains circulating consecutively between January and June 2022 (Fig. S2). The final sample consisted of 194 individuals who were later infected by either the BA.1 ($N = 153$) or BA.2 ($N = 41$) strains, and 151 individuals who remained uninfected. Study participants had a median age of 18 years (IQR 9–40), with a majority being female (211/345 [61%]). By the time of sampling, most participants experienced at least one prior SARS-CoV-2 infection (322/345 [93%]) and had received at least one vaccine dose (233/345 [68%]) (Table 1). The average number of prior vaccine doses among individuals sampled before BA.2 introduction compared to those sampled before BA.1.

We measured serum neutralization of BA.1, BA.2, and D614G SARS-CoV-2 using pseudovirus assays, and assessed Spike-binding titers using ELISA. Samples were collected prior to infection or up to 4 days after the first RT-PCR-positive detection within a household with a median of −4 days (Interquartile range [IQR] −17−0) relative to the first positive detection date for the neutralization assays, and −6 days (IQR −60−0) for Spike differing by sample availability and with 84% samples

shared across assays (Table 1). We found that all measured immune markers were consistently higher before the BA.2 introduction compared to the BA.1 introduction (Table 1). While all markers measured in the study were co-linear at the time of sampling, the greatest collinearity was observed between BA.1- and BA.2-specific titers (Fig. S4). We then evaluated how these markers correlate with outcomes. In our univariate analyses, neutralizing titers across viruses were significantly associated with protection from infection with BA.2 but not BA.1 (Table S1, Fig. S5). In contrast, neutralizing titers were significantly associated with protection from symptomatic infections with BA.1 and BA.2 infections, except for D614G titers in relation to symptomatic BA.1 infection ($P$-value 0.106) (Table S2, Fig. S6).

Similarly, after adjusting for age, infection history and vaccine history, we found that a four-fold linear increase in neutralizing titers across neutralization assays was significantly associated with protection from BA.2 infection–corresponding to 45% protection (95%

**Table 1 | Participant characteristics**

| Characteristic | Overall | BA.1 wave | BA.2 wave |
|---|---|---|---|
| Participants – N | 345 | 251 | 94 |
| Households – N | 91[a] | 66 | 32 |
| Participants per household – Median (IQR) | 3 (2, 5) | 3 (2, 5) | 3 (2, 3) |
| Age – Median (IQR) | 18 (9, 40) | 18 (9, 39) | 17 (10, 41) |
| Female sex – N (%) | 211 (61%) | 151 (60%) | 60 (64%) |
| Infected – N (%) | 194 (56%) | 153 (61%) | 41 (44%) |
| Infection severity – N (%) | | | |
| Subclinical | 89 (46%) | 77 (50%) | 12 (29%) |
| Mild | 78 (40%) | 52 (34%) | 26 (63%) |
| Moderate | 24 (12%) | 21 (14%) | 3 (7.3%) |
| Severe | 3 (1.5%) | 3 (2.0%) | 0 (0%) |
| Infection history – N (%) | | | |
| No prior infections | 23 (6.7%) | 18 (7.2%) | 5 (5.3%) |
| 1 prior infection | 235 (68%) | 171 (68%) | 64 (68%) |
| 2 prior infections | 80 (23%) | 58 (23%) | 22 (23%) |
| 3 prior infections | 7 (2.0%) | 4 (1.6%) | 3 (3.2%) |
| Time since last infection – Median (IQR) | 406 (163, 584) | 380 (150, 580) | 437 (263, 689) |
| Vaccination history | | | |
| Unvaccinated | 112 (32%) | 90 (36%) | 22 (23%) |
| 1 prior dose | 71 (21%) | 59 (24%) | 12 (13%) |
| 2 prior doses | 75 (22%) | 57 (23%) | 18 (19%) |
| >2 prior doses | 87 (25%) | 45 (18%) | 42 (45%) |
| Time since last Vax dose – Median (IQR) | 77 (34, 102) | 62 (22, 93) | 112 (77, 162) |
| Pre-infection titers – Median (IQR) | | | |
| Neut BA.1 ID$_{50}$ | 1347 (180, 3330) | 1061 (146, 3130) | 1839 (554, 3943) |
| Neut BA.2 ID$_{50}$ | 1482 (292, 4517) | 1332 (218, 3975) | 2150 (709, 5338) |
| Neut DG14G ID$_{50}$ | 5038 (1315, 11694) | 4798 (1173, 12509) | 5132 (1954, 10882) |
| Spike ELISA titer | 8553 (2198, 28937) | 7166 (1505, 26138)[b] | 10660 (5168, 36256) |
| Sample time before infection (Neut)[c] – Median (IQR) | −4 (−17, 0) | −4 (−14, 0) | −3 (−22, 0) |
| Sample time before infection (Spike)[c] – Median (IQR) | −6 (−60, 0) | −6 (−67, 0)[b] | −6 (−60, 0) |

[a]91 unique households. 7 households were activated in both waves.
[b]1 individual with unknown titer.
[c]Among BA.1 or BA.2 infected participants; "Neut": samples for neutralizing titers; "Spike": for spike-binding.

Confidence Interval [CI] 21–64%) for BA.1 neutralizing titers, 45% (CI 21–64%) for BA.2 neutralizing titers, and 51% (CI 23–71%) for D614G neutralizing titers–but not with protection from BA.1 infection (Fig. S7, Table S3). Protection is interpreted as the relative reduction in the odds of infection per four-fold increase in antibody titers, calculated as 1 − OR x 100. Spike-binding titers were not associated with protection from BA.1 or BA.2 infection (Fig. S7, Table S3). However, both neutralizing and spike-binding titers were associated with protection from symptomatic infection across waves (Fig. S7, Table S4).

A four-fold increase in neutralizing titers was also significantly associated with protection from symptomatic BA.1 infection: 28% protection (95% Confidence Interval [CI] 12–42%) for BA.1 neutralizing titers, 28% (CI 12–42%) for BA.2 neutralizing titers, and 28% (CI 7–44%) for D614G neutralizing titers. For symptomatic BA.2 infection, a four-fold increase in neutralizing titers was associated with 47% protection CI 24–65%) for BA.1 neutralizing titers, 43% (CI 20–62%) for BA.2 neutralizing titers, and 57% (CI 29–75%) for D614G neutralizing titers. Although similar levels of protection were observed for each four-fold increase in neutralizing titers across assays, the average pre-exposure titer values were higher for D614G than for the Omicron subvariants (Fig. S7, Table 1). Results did not differ after adjusting for household clustering in this household design and time since last exposure. The estimated within-household correlation parameters were low, suggesting that variation in infection risk was driven primarily by individual-level rather than household-level effects (Tables S5–8).

We then analyzed infections and symptomatic infections in the range of observable titers among vaccinated and previously infected adults to derive protection thresholds for BA.1 and BA.2 infection outcomes. Titers to achieve 50% and 80% protection from BA.2 infections were similar for BA.1 and BA.2 neutralizing titers. However, achieving the same level of protection using D614G neutralizing titers required more than three times higher than BA.1 or BA.2 neutralizing titers (Fig. 1A, Table S9–10). In contrast, titers required for 50% or 80% protection from symptomatic BA.1 or BA.2 infection were lower than those needed to prevent BA.2 infection. Similarly, protective titers against contemporaneous strains were lower compared to D614G neutralizing titers (Fig. 1B, Table S9-10). For instance, achieving 50% protection from BA.1 infection required a BA.1 neutralizing titer of 39 and BA.2 neutralizing of 53, whereas a D614G neutralizing titer of 222 was needed. Protective titers based on Spike-binding assays were also reported (Figure S8).

### Neutralizing antibodies mediate protection conferred by vaccination or prior infection

We next employed causal mediation analysis to evaluate the role of antibody responses in mediating the protective effects of prior infection or vaccination against subsequent infection and symptomatic infection. For these analyses, we combined both Omicron waves and adjusted for Omicron wave, age and prior infection or vaccination (as main exposure or a confounder, as appropriate). "Homotypic" antibodies (e.g., BA.1- or BA.2-neutralizing antibodies against BA.1 or BA.2 infections, respectively) significantly mediated protection, with mediation effect estimates indicating that every four-fold increase in titer mediated 11% (CI 4–21%, $p = 0.006$) and 12% (CI 6–24%, $p = 0.002$) protection from prior infection for infection and symptomatic infection, respectively (Fig. 2, Table S11). Additionally, for every four-fold increase in titer, "homotypic" neutralizing antibodies significantly mediated 5% (CI 2–9%, $p < 0.001$) protection against infection and 7% (CI 4–11%, $p < 0.001$) protection against symptomatic infection from vaccination. D614G neutralizing antibodies also significantly mediated protection from prior infection and vaccination, though with slightly lower mediation effects. In contrast, D614G spike-binding antibodies did not significantly mediate protection against infection and only mediated significant protection against symptomatic infection from both vaccination and prior infection (Fig. 2, Table S11).

## Discussion

In this carefully monitored cohort of household participants in Managua, Nicaragua, we found that both homotypic Omicron-neutralizing antibody titers (e.g., anti-BA.1 titer for BA.1 wave participants) and ancestral SARS-CoV-2-neutralizing antibody titers were informative correlates of protection against symptomatic infection during Omicron BA.1 and BA.2 infection waves in 2022. Mediation analysis further revealed that neutralizing antibodies causally mediate protection from infection conferred by vaccination or prior infection. Our findings confirm that neutralizing antibody titers serve as a meaningful correlate of protection, measuring a causally protective factor, that may inform risk assessment.

Many prior studies have demonstrated, to varying degrees, that humoral immunity serves as a correlate of protection from SARS-CoV-2 infection or severe symptoms[2–7,9,15,17–24]. The results presented here align with previous findings on ancestral SARS-CoV-2-neutralizing titers but also extend beyond prior work in key ways. First, the participants in this study reside in Managua, Nicaragua, where SARS-CoV-2 had circulated extensively before the Omicron waves. The original SARS-CoV-2 strain started circulation in 2020, by 2021 Gamma and Delta predominated, and starting 2022 Omicron took over, first BA.1 and then BA.2 quickly replaced BA.1[25]. By the time of this study's sampling, people had been infected multiple times and subsequently vaccinated[16]. The vaccines administered in this population included Sputnik and AstraZeneca for adults, and Abdala and Soberana for children[26].

To the best of our knowledge, no prior studies on neutralizing antibody correlates of protection from COVID-19 have focused on participants in Nicaragua or any other Central American countries. One prior study from South America, which included participants from Chile and Peru, analyzed ancestral-neutralizing titers in a phase 3 trial of AZD1222 (ChAdOx1 nCoV-19) vaccine[27]. Our study provides critical insights into protective immunity in a region that has been underrepresented in global SARS-CoV-2 research. Here, we considered Nicaraguan participants with a variety of protective exposure histories (prior infections and various vaccines) and provided mediation analysis results separated by exposure.

Second, we utilized a household-based cohort study with an embedded transmission study design triggered by known virus exposures. Compared to clinical trials or standard observational cohort studies, exposures to infected individuals are expected to be more consistent across participants in household studies and more closely temporally linked to the measurement of antibody titers. For example, in Sun et al., samples were collected several weeks before high incidences of Omicron infections[15]. Therefore, the results herein may be more reflective of similar viral exposure challenges across participants than in prior work, and correlates of protection from this study may be useful when considering protection levels at the time of exposure.

Third, we measured titers against the ancestral virus, to which participants were originally exposed earlier in the pandemic and/or by vaccination, as well as against contemporaneous Omicron subvariants BA.1 and BA.2. Relatively few correlates of protection studies have been published with such concurrent analyses of neutralizing antibodies directed against different prior and contemporaneous variants, and none have been household-based cohort with an embedded transmission study triggered by defined index case exposures around the time of titer measurement like this study[15,22,23,28,29]. Recent COVAIL trial reports similarly identified neutralizing antibodies as correlates of protection against Omicron across mRNA and recombinant protein vaccines[22,23]. Our household-based study extends these findings to a community setting where heterogeneous exposures, including multiple infections and vaccines less frequently evaluated globally. One household-based longitudinal cohort study of unvaccinated people in South Africa, without the nested triggering of sample collection that

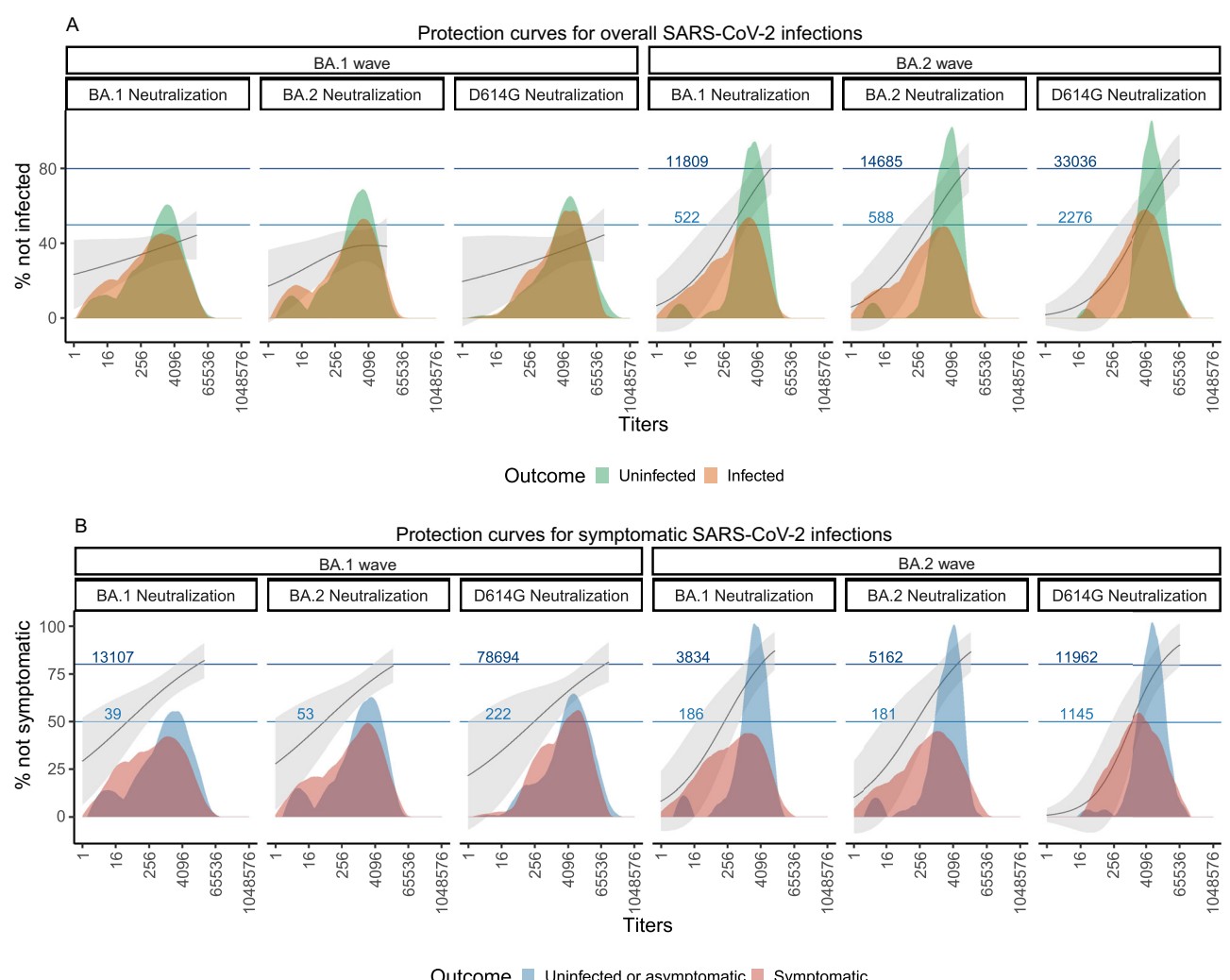

**Fig. 1 | Neutralizing titers as a correlate of protection by Omicron wave and assay.** Relationship between neutralizing titers ($ID_{50}$) and protection against BA.1 and BA.2 infection (**A**) and symptomatic infection (**B**) during their respective waves. The solid lines represent the mean predicted protection curves, calculated as 1 - predicted risk, according to a Generalized Additive Model using a Gaussian distribution (see Methods), with the shaded ribbon bands denoting the 95% confidence intervals (CIs) around the mean predicted protection estimates, derived from vaccinated and prior-infected adults. The horizontal blue solid lines mark the points where the predicted protection curves cross the 50% and 80% protection thresholds, highlighting the titer levels required to achieve these levels of protection. Embedded within the figure are histograms showing the raw distribution of neutralization titers, stratified by outcome (infection vs. no infection, symptomatic vs. asymptomatic). Data were derived from all 345 study participants stratified by Omicron wave and outcome: BA.1 wave ($n = 251$; 143 infected vs. 108 uninfected, 105 symptomatic vs. 146 not symptomatic) and BA.2 wave ($n = 94$; 41 infected vs. 53 uninfected, 29 symptomatic vs. 65 not symptomatic), with one pre-exposure sample per participant. Neutralizing titers ($ID_{50}$) were obtained by averaging inhibition values at each dilution across technical triplicates of each sample and fitting a 5-parameter log-logistic model (see Methods).

we deployed, measured anti-D614G and anti-BA.1 neutralizing titers months prior to a BA.1 infection wave. In that study, the authors found that BA.1-neutralizing titers mediated protection from infection elicited by prior infections, but also that the difference in neutralizing titer between D614G and BA.1 was itself no longer a correlate of protection, suggesting that homotypic-neutralizing antibody components of ancestral-neutralizing titers are the basis of the nAb correlate of protection[15]. Our results showed that protective levels of D614G-neutralizing antibodies are likely higher than protective levels of homotypic-neutralizing antibodies, concordant with the results of Sun and colleagues, in both unvaccinated and vaccinated individuals, and supporting our hypothesis that variant-specific titers would be better correlates of protection. Intriguingly, our GLM-based correlate of protection analysis found that 4-fold increases in ancestral- and homotypic-neutralizing antibodies provided essentially equivalent increases in protection, despite higher average titers across the cohort directed against the ancestral strain. This result may also support the

notion that during BA.1 and BA.2 waves, a fraction of cross-reactive ancestral-neutralizing antibodies also had activity against Omicron subvariants. Further, this result points to ancestral-neutralizing antibody titer being a useful correlate of protection even for a viral challenge with later Omicron BA.1 or BA.2 variants.

Our results are also consistent with literature showing that neutralizing antibodies causally mediate protection from infection by SARS-CoV-2 conferred by prior exposures. Here, we further show that neutralizing antibodies mediate protection against infection, and more dramatically, protection against symptomatic infection, conferred by vaccination or prior infection. The cohort studied herein had exposure to global vaccines less frequently studied and most individuals had one or more infections before vaccination.

Finally, we show that serum antibodies, in particular neutralizing antibodies, are clearly informative correlates of protection from symptomatic infection. However, they were correlated with protection from asymptomatic infections only at higher titers, as detected by

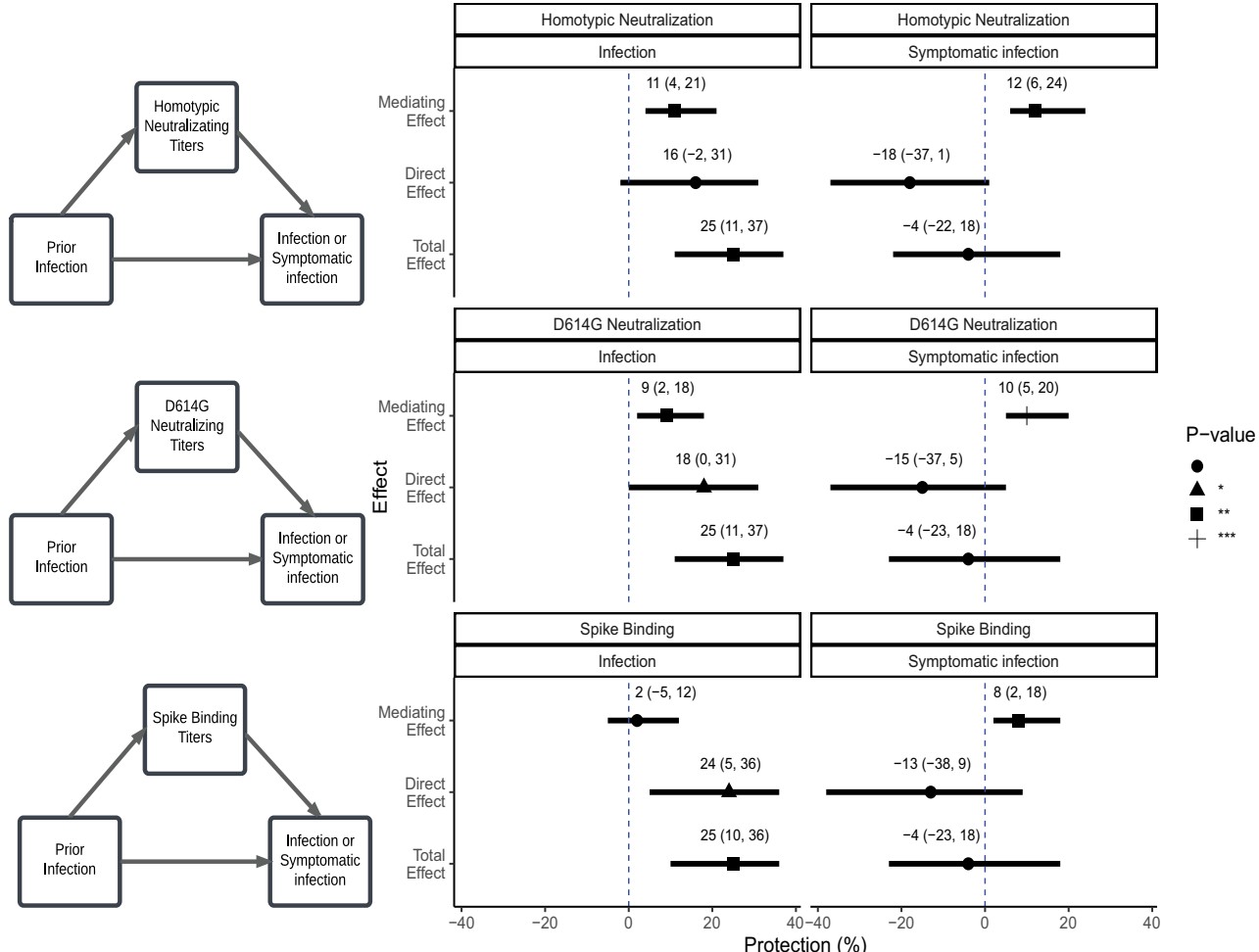

**Fig. 2 | Mediation analysis of antibody effects on infection and symptomatic infection through prior infection.** The first column depicts the diagrams of the mediation framework analysis. The second column presents the corresponding estimates for each antibody measure (homotypic-neutralizing, D614G-neutralizing, and spike-binding antibodies), displaying the total effect, average direct effect, and average causal mediating effect from any prior infection on protection against infection and symptomatic infection. Each mediation analysis was adjusted for age and prior vaccination. Protection effects are expressed as percentages ([1-RR] x100). Point estimates (means) are indicated by symbols (circle, triangle, square, or cross), with solid horizontal lines representing the 95% confidence intervals (CIs). Dashed line indicates null effect. Mediation effects were estimated using Poisson regression models, with statistical significance assessed via non-parametric bootstrapping with 1000 simulations and accelerated bias-corrected (BCa) 95% confidence intervals (two-sided tests). *P*-values are indicated for statistical significance, with ***, and * corresponding to *P*-values of <0.001, <0.010, and <0.050, respectively; no adjustments for multiple comparisons were applied; exact *P*-values are shown in Table S11. Data were derived from all 345 study participants (237 infected vs 108 uninfected, 134 symptomatic vs 211 not symptomatic), with one pre-exposure sample per participant. Neutralizing titers ($ID_{50}$) were obtained by averaging inhibition values at each dilution across technical triplicates of each sample and fitting a 5-parameter log-logistic model (see Methods).

multiple prospective peri-exposure nucleic acid amplification tests paired with symptom questionnaires. This finding aligns with the pathophysiology of COVID-19 and the known limitations of serum antibodies, as mucosal immunity plays a key role in protection against initial infection[30]. Thus, while serum antibody measurements are valuable for predicting protection against symptomatic COVID-19, they do not equally correlate with sterilizing immunity against overall SARS-CoV-2 infection.

This study has several limitations. First, there is potential ascertainment bias in our household-based cohort study design, as households with higher susceptibility to viral transmission and disease may have had a greater likelihood of being included due to the occurrence of an index case. Second, while we utilized neutralizing antibody assays to establish protective levels, variations in assay methodologies across different clinical laboratories may influence the exact protective thresholds reported, limiting cross-study comparisons. Third, our analysis to detect correlates of protection for BA.1 infection may be underpowered. Fourth, there may be a difference in the number of unreported SARS-CoV-2 exposures for participants in the earlier BA.1 and later BA.2 waves, which could affect the difference in correlation with protection during these two waves. Fifth, the assumption in our analyses that all household members experienced equivalent exposure to the virus may not fully capture variability in individual exposure risks within households. Sixth, potential confounders such as health-seeking behavior and risk avoidance, comorbidities, host genetic susceptibility, and socioeconomic status (SES) may influence both prior infection or vaccination (the exposure), antibody levels (the mediator), and the risk of infection (the outcome), but were not fully captured in our analysis. Finally, while our causal mediation analysis supports the role of neutralizing antibodies in mediating protection, several key assumptions of the mediation framework, including exchangeability, consistency, and positivity, may not be fully satisfied in this observational context. In particular, the positivity assumption may be violated due to the very low prevalence of individuals without prior SARS-CoV-2 infection in this cohort. This lack of adequate representation of uninfected participants reduces overlap between exposure groups, which

can result in unstable or biased mediation effect estimates. Therefore, causal interpretations of these results should be made cautiously.

In conclusion, in this household-based cohort with an embedded transmission study conducted in Managua, Nicaragua, we demonstrated that neutralizing antibody titers, particularly against future strains, are informative correlates of protection from infection and symptomatic SARS-CoV-2 infection. Mediation analyses provide evidence that neutralizing antibodies contribute to protection conferred by vaccination or prior infection, underscoring their role as a mechanistic correlate of immunity. Our findings emphasize the need for continuous surveillance of immune responses to evolving viral variants.

## Methods

### Study design and participants
The Household Influenza Cohort Study (HICS) is an ongoing prospective cohort study in District II of Managua, Nicaragua in the Health Center Socrates Flores Vivas (HCSFV). Established in 2017 to investigate influenza, it was expanded in early 2020 to include SARS-CoV-2 infection and disease[16]. All cohort participants undergo regular blood sampling–twice per year in 2020 and 2021, and then annually thereafter. At the first sign of any illness, participants are instructed to seek care at the study health clinic, where they receive primary care and diagnostic testing (Fig. S1).

An embedded household transmission study is activated once a participating household member is confirmed positive for SARS-CoV-2 (Fig. S1). After activation, study personnel visit the household on days 0, 3, 7, 14, 21, and 30 to collect combined nasal/oropharyngeal swabs, maintain symptom diaries, and test all household members regardless of symptoms. Serum samples are collected on the same day as activation. Active surveillance of household contacts allows the detection of symptomatic and asymptomatic SARS-CoV-2 infections.

These studies received approval from the institutional review boards of the Nicaraguan Ministry of Health and the University of Michigan (HUM00119145 and HUM00178355). Written informed consent or parental permission was obtained for all enrolled participants, and children aged 6 years or older also provided assent.

### Inclusion and exclusion criteria
Households were eligible for inclusion if they experienced an introduction of either BA.1 or BA.2 SARS-CoV-2 strains, which circulated sequentially from January and June 2022 (Fig. S2). Pre-exposure samples were included in the analysis if they were collected from 10 days before up to 4 days after the household's activation date. If a participant lacked a sample within this window but had one collected within the proceeding 90 days- without documented SARS-CoV-2 infection or vaccination in the interim- and before the first positive respiratory sample in the activation period, it was also accepted as a pre-exposure sample. Based on these criteria, 14 participants were excluded.

Strain ascertainment was performed by direct sequencing from household members or by single imputation using weekly sequencing data from the Ministry of Health of Nicaragua (Managua department) (Fig. S4). Two households (totaling 11 individuals) were subsequently excluded because they lacked sufficient data for direct sequencing or single imputation (multiple strains were circulating that week). In total, these criteria resulted in a final cohort of 345 participants, 251 within the BA.1 Omicron wave and 94 individuals within the BA.2 Omicron wave.

### Laboratory methods
Blood samples were analyzed in pairs (current vs. baseline) using enzyme-linked immunosorbent assay (ELISA) protocols adapted from the Krammer laboratory for the detection of antibodies against SARS-CoV-2 spike receptor binding domain (RBD), Spike and Nucleoprotein (NP). The RBD and spike proteins used in these assays were produced

in single batches at the Life Sciences Institute at the University of Michigan, based on the original SARS-CoV-2 strain. Due to its specificity, RBD was used for initial screening (positive/negative), while positive samples were subsequently titrated using the spike assay. The NP ELISA was also performed to differentiate vaccine-induced responses from those due to infection. Real-time reverse-transcription polymerase chain reaction (RT-PCR) was conducted following the protocol described by Chu et al.[31].

Neutralization titers against BA.1, BA.2 and D614G SARS-CoV-2 strains were measured using vesicular stomatitis virus–based pseudoviruses (VSV) bearing different SARS-CoV-2 spike proteins produced in HEK293T cells, as previously described[11]. After transfecting spike-encoding plasmids, a VSV-G pseudotyped ΔG-luciferase was used to generate the pseudoviruses, which were harvested and standardized by determining the 50% tissue culture infectious dose ($TCID_{50}$) in Vero-E6 cells. Heat-inactivated serum samples were then serially diluted, incubated with each pseudovirus, and added to Vero-E6 cells, followed by a 16-hour incubation. Luciferase activity was subsequently measured, and $ID_{50}$ values for each strain-specific pseudovirus were calculated using a five-parameter log-logistic model to compare neutralization responses across the different SARS-CoV-2 variants. Additional details are included in the supplementary material.

### Endpoints
Symptoms reported in daily symptom diaries and during clinic visits were used to categorize illness severity into four levels. "Severe" cases included individuals requiring hospitalization, and those deemed in need of hospitalization by clinicians. No deaths due to COVID-19 were recorded during the Omicron BA.1 or BA.2 waves in the HICS study. "Moderate" cases encompassed individuals presenting with difficulty breathing, shortness of breath, rapid breathing, tight chest feeling, chest pain, ALRI, SARI, crepitus, chest wall indrawing, rhonchi, wheezing, or an overall poor condition. "Mild" cases were defined as those with loss of smell/taste, fever, or at least two other symptoms. Finally, "subclinical" cases involved no more than one symptom that did not meet the criteria for higher severity. The primary endpoints of this study were the percentage protection against RT-PCR-confirmed infection–including subclinical, mild, moderate, and severe infections, (hereafter referred to as 'infection')–and symptomatic infection (including mild, moderate, and severe infections) among households with SARS-CoV-2 BA.1 or BA.2 exposure.

### Independent variables
Antibody titers (the primary independent variables of interest) across assays were log-transformed using base 4 ($\log_4$[titer]). Assays included D614G, BA.1 and BA.2 neutralization tests, and Spike ELISA. Dilution factors leading to 50% neutralization ($ID_{50}$s) were calculated using a 5-parameter log-logistic model among the neutralization assays. A modification of the Reed and Muench formula was used to calculate titers for the Spike ELISA[32]. Confounders included age (continuous variable), vaccination status (any prior vaccination, dichotomous variable) and infection history (any prior infection, dichotomous variable). Infection history and vaccination ascertainment details are included as supplementary materials. The selection of confounders was based on directed acyclic graph (DAG) analysis supported by the existing literature (Figure S3).

### Statistical analysis
Univariate analyses using Wilcoxon rank sum test were applied to examine the relationship between antibody titer and infection outcome by assay and Omicron wave (BA.1 or BA.2). We also assessed the multivariate association between titers and infection outcomes by Omicron wave and assay with generalized linear models (GLMs) under a binomial family. We tested the assumption of independence by adjusting for household IDs, and the linearity of the odds ratios by

comparing GAM and GLM outputs. Two sensitivity analyses were added to assess the robustness of the multivariable findings. First, we applied a Generalized Estimating Equations (GEE) framework with an exchangeable working correlation structure, clustering on household ID. Robust (sandwich) standard errors and 95% confidence intervals were extracted to account for within-household correlation. Second, we extended the multivariable model to additionally adjust for time since last exposure (infection or vaccination), which may influence both antibody levels and risk of reinfection.

Protection associated with increasing titer levels was quantified as 1 − odds ratio (OR) x 100 to quantify the relative reduction in odds of infection or symptomatic infection per a 4-fold increase in the antibody marker, holding covariates fixed, similar to calculations of vaccine effectiveness. We also determined protective cut-offs or thresholds to achieve 50 and 80% protection (reduction in risk) using multivariate GLM and Generalized Additive Models (GAM) with thin-plate spline regression with three knots to allow for non-linear effects. Multivariate GAM and GLM models were adjusted for the confounders mentioned above. Protective thresholds were estimated by holding covariates constant, specifically representing vaccinated adults with prior natural infection, who for the threshold analysis constituted our primary population of inference.

We also conducted a causal mediation analysis, as a secondary analysis, to examine how neutralizing antibodies mediate protection from prior vaccination and natural infection against both overall and symptomatic infection. To avoid the non-collapsibility property of logistic regression, we used log-binomial regressions to estimate relative risks (RR), and calculated protection as 1-RR x 100. Analyses were conducted within a GLM framework with non-parametric bootstrapping and accelerated bias-corrected 95% confidence intervals applying 1000 simulations. For these analyses, we combined both Omicron waves and adjusted for Omicron wave, age and prior infection or vaccination (as main exposure or a confounder, as appropriate). That is, when prior infection was the exposure, vaccination was considered a confounder, and vice versa.

All analyses examining the likelihood of infection or symptomatic infections included the computation of 95% confidence intervals (CIs) using standard errors, assuming a normal distribution with an α level of 0.05, except for the mediation analyses. Statistical significance was evaluated using two-tailed Wald tests on the regression coefficients. All statistical analyses were conducted using R (R Foundation for Statistical Computing v4.3.1) and mediation statistical package (v4.5.0)[33,34].

### Reporting summary
Further information on research design is available in the Nature Portfolio Reporting Summary linked to this article.

## Data availability
Researchers interested in accessing the study data are encouraged to submit a formal request to A.G. or the Committee for the Protection of Human Subjects at the University of Michigan. To uphold ethical standards and ensure appropriate data use, each request will undergo a case-by-case review and approval process. Additionally, as the data include information collected in Nicaragua, access is subject to Nicaraguan data ownership regulations and may require approval from relevant Nicaraguan authorities. Final approval is expected within two months. Materials requests and correspondence should be directed to A.G. (gordonal@umich.edu) and D.D.H. (dh2994@cumc.columbia.edu). Accession codes from sequenced participants in the study can be found in Table S12.

## Code availability
Code required to reproduce these results, including a dummy dataset to run the code are publicly accessible at https://zenodo.org/records/16804602.

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

## Acknowledgements

We thank the study personnel at the Centro de Salud Sócrates Flores Vivas, the Nicaraguan National Virology Laboratory, and the Sustainable Sciences Institute in Nicaragua. We are particularly grateful to the study participants and their families. This work was supported by the National Institute of Allergy and Infectious Diseases at the National Institutes of Health (award no. R01 AI120997 to A.G. and contract nos. HHSN272201400006C and 75N93021C00016 to A.G.), as well as by a grant from the Open Philanthropy Project, NIH SARS-CoV-2 Assessment of Viral Evolution (SAVE) Program (subcontract no. 0258-A700-4609 under federal contract no. 75N93021C00014) to D.D.H., and the Gates Foundation (project INV019355) to D.D.H.; A.G. is supported by the Biosciences Initiative at the University of Michigan through a Mid-career Biosciences Faculty Achievement Award. The funders of the study had no role in study design, data collection, data analysis, data interpretation, or writing of the report.

## Author contributions

J.V.Z.: conceptualization, data curation, formal analysis, methodology, software, validation, visualization, writing—original draft, writing—review and editing. I.A.M: investigation, formal analysis, methodology, writing—original draft, writing—review and editing. A.S.: data curation, methodology, writing—review and editing. H.E.M: data curation, methodology, writing—review and editing. M.P. and G.K: investigation, project administration, writing—review and editing. Y.S., R.L., G.V., S.O., N.S., Q.W., and L.L.: investigation, writing—review and editing. I.G.: data curation, literature search, investigation. A.B.: investigation, methodology, supervision, project administration, writing—review and editing. D.D.H: conceptualization, investigation, methodology, project administration, resources, supervision, writing—review and editing. A.G.: conceptualization, funding acquisition, investigation, methodology, project administration, resources, supervision, writing—review and editing.

## Competing interests

A.G. has received institutional funding from Flu Lab and Open Philanthropy; personal honoraria from Hope College and the La Jolla Institute of Immunology; compensation for expert testimony from Berman and Simmons; and travel support from the Gates Foundation and the National Institutes of Health (NIH). A.G. has also served or serves in an advisory capacity to Janssen Pharmaceuticals and Sanofi Pasteur. D.D.H. co-founded TaiMed Biologics and RenBio, and he serves as a consultant for WuXi Biologics and Brii Biosciences and is a board director at Vicarious Surgical. All other authors declare no potential competing interests.

## Additional information

¹Department of Epidemiology, University of Michigan, Ann Arbor, MI, USA. ²Sustainable Sciences Institute, Managua, Nicaragua. ³Aaron Diamond AIDS Research Center, Vagelos College of Physicians and Surgeons, Columbia University Irving Medical Center, New York, NY, USA. ⁴Department of Pathology and Cell Biology, Vagelos College of Physicians and Surgeons, Columbia University Irving Medical Center, New York, NY, USA. ⁵New York Blood Center, New York, NY, USA. ⁶Laboratorio Nacional de Virología, Centro Nacional de Diagnóstico y Referencia, Ministerio de Salud, Managua, Nicaragua. ⁷Centro de Salud Sócrates Flores Vivas, Ministerio de Salud, Managua, Nicaragua. ⁸These authors contributed equally: José Victor Zambrana, Ian A. Mellis. ✉e-mail: dh2994@cumc.columbia.edu; gordonal@umich.edu

