## [Transparent Peer Review file · Nature Communications]

Variant-specific antibody correlates of protection against SARS-CoV-2 Omicron symptomatic and overall infections

Corresponding Author: Professor Aubree Gordon

Version 0:

Reviewer comments:

Reviewer #1

(Remarks to the Author)

The study provides insights into the correlates of protection against SARS-CoV-2 Omicron variants, focusing on a population in Nicaragua which is an underrepresented region in COVID-19 research. The use of a household-based cohort study with an embedded transmission study design is particularly interesting, allowing for the tracking of virus exposure and antibody responses in a cohort that may have more consistent exposures to the virus than those in other studies. The mediation analysis is novel and attempts to assess the causal role of neutralizing antibodies in protection, rather than just as a correlate. I congratulate the authors on an interesting and well-thought-out study.

Despite this, the general findings of the study are not particularly novel. The correlates with neutralizing antibodies are established, and it is expected that variant-specific antibodies would also be correlates, given the correlated nature of ancestral and variant-specific correlates (as shown in this article as well). Given the rapidly evolving variant landscape, the authors do not demonstrate how variant-specific correlates could be used to improve clinical practice or public health policy in practice. This limits the broader impact of this study. While the Central America focus is important to build out the understanding among all populations, the findings are mostly confirmatory, rather than creating new insight.

The findings that are novel are those presented in the mediation analysis, where the authors aim to delve further into a causal relationship, rather than just a correlated one. However, the authors also rightfully acknowledge that this analysis carries some assumptions that limit the causal interpretations. It would be helpful if the authors elaborated on these assumptions and common ways they may be violated to aid the reader in interpreting their robustness. An example that comes to mind is whether there are other unobserved confounders between the pre-exposure titre and the outcome. This could reasonably be comorbidities, other measures of immune function, or genetic factors (given this is a household study). Any such unobserved confounders here would introduce biasing paths, which might be helpful to illustrate with further DAGs like that in Figure S3.

Finally, there are some methodological concerns that need addressing for the GLM and GEM analysis. Figures S5 – S7 show that homogeneity assumptions are likely to be violated as the variance of titres is much larger for those infected than those not infected. This is fine for the univariate analysis, which is non-parametric, but there is no discussion on how this potential violation to assumptions is addressed or impacts the multivariable analysis, which are the primary analysis outcomes presented in the abstract.

Minor Comments:

- Numbers for protection against infection for BA.2 in the Abstract do not match the numbers in the results or Table S3. They are close, so one of these is likely a typographical error.
- In Figure S3, the prior vaccinations may also have a causal relationship with prior infections. I note this doesn't impact biasing paths and so is not consequential, but may be worth mentioning in the figure legend.
- The Nature communications form on reporting summary on statistics appears incorrect: No discussion on tests of

assumptions for GLM and GEM analysis – though this is ticked as “confirmed”; The description of statistical parameters is not complete. Means and CIs are presented, but not standard deviations.; P-values are presented, but not their associated test statistics and degrees of freedom (for GLM/GEM analysis).

(Remarks on code availability)

Code was unavailable to be reviewed as it was not part of the reviewing pack, and does not appear to be publicly available.

Reviewer #2

(Remarks to the Author)

The dataset reported this manuscript is unique and provides a valuable opportunity to study the relationship between SARS-CoV2-specific humoral immune responses and risk of COVID.

Major comments:

1. The statistical results for the two endpoints, symptomatic and asymptomatic infections, are qualitatively the same, as can be seen from comparing panel A and B in Figure 1. It could make a clearer presentation if the main text focuses on one set of endpoints and the supplementary results cover the other.
2. Authors defined 'protection' by $1 - OR$, where OR is the coefficient of the titer level in a GLM with a binary outcome (infection or not). The language is confusing. It is perhaps clearer to report it as association.
3. On the first objective, the main results are that the antibodies are not correlates in the BA.1 wave but are correlates in the BA.2 wave. This is an interesting and intriguing pair of results. We find the description of these two results unclear and the discussion of the results lacking. Is it true that we can think of it as two separate datasets close in time? Would the two datasets be from the same population, i.e. wouldn't the BA.1 wave change the immunity of the population? For each dataset, should the household be treated as a cluster in regression? How do we interpret the coefficients for antibody markers when infection history and vaccine history, which affect immune biomarker levels, are adjusted in regression?
4. On the second objective, the mediation analysis, the presentation is very unclear, which makes the results difficult to interpret. A reference example may be the mediation section in <https://www.science.org/doi/10.1126/science.abm3425>. Some specific comments: First, is prior infection and prior vaccination treated separately or together? Second, in the cohort, according to Table 1, there are only 23 (6.7%) ppts with no prior infection. How many got infected among these 23 ppts? If the authors thought of 'prior infection' as a treatment, the treatment is very imbalanced. Third, time since last infection is heterogeneous and this implies the 'direct effect' from prior infection on infection is heterogeneous. Not sure if authors considered this in the mediation analysis. Fourth, mediation analysis relies on a 'positivity assumption' of the mediator. Again, this is likely not satisfied. Put together, I do not find the proportion mediated reported by the authors convincing.

Minor comments:

1. The literature research lacks references to some more recent (2023-present) papers on Omicron correlates.
2. Authors said they determined the 'threshold' (see methods) using splines. The supplementary tables define what threshold means. That detail should be moved to the methods.

(Remarks on code availability)

Reviewer #3

(Remarks to the Author)

(Remarks on code availability)

Version 1:

Reviewer comments:

Reviewer #1

(Remarks to the Author)

The authors have satisfactorily addressed my comments from the prior review and I have no further comments, apart from comments on code availability.

(Remarks on code availability)

The submitted code comprises approximately 1,500 lines. While a full line-by-line review was not feasible, a general assessment indicates that the code is well-organized, clearly formatted, and adequately commented. These attributes suggest that the code should be understandable to readers with a working knowledge of R. Spot checks did not reveal any coding errors or issues.

The availability of the code significantly supports the reproducibility of the methods described in the manuscript. To further enhance reproducibility, it is recommended that the authors provide dummy datasets compatible with the code. This would allow reviewers and readers to compile and validate the code independently. For full reproducibility, access to the original datasets used in the study would be necessary. However this depends on whether the data can be made publicly available.

Reviewer #2

(Remarks to the Author)

We appreciate the authors' detailed responses. Some remaining concerns are listed below.

1. The authors' response did not cite any references to support their claim that the use of the term "protection" for 1-OR is widely accepted in today's literature. I looked up the three new references the authors added. None used protection to refer to 1-OR: Marking et al. reported RR, Zhong et al. reported area under ROC, Canetti et al. reported OR. Moreover, the x axis and the y axis labels in Fig 1 do not work together. For example, the first panel seems to indicate that the protection is around 25% when the BA.1 neut titer is 1, which does not make much sense. A similar plot in Canette et al. can be used as an example of a truer representation of the results. Canette et al. Fig 2B labels the y axis as OR and the caption reads: "Estimated odds ratio of infection as a function of antibody levels compared to an individual with undetectable antibody levels, based on a crude logistic regression model." The key here is that the OR is a contrast and it is associated with a contrast in titer, not in absolute titer.

2. Violation of the positivity assumption in a causal mediation analysis typically means the probability of the mediator taking any level conditional on the baseline covariates bounded away from 0; see, e.g., Identification, Inference and Sensitivity Analysis for Causal Mediation Effects. For instance, depending on baseline covariates like previous vaccine and infection history, the mediator level may be poorly overlapped and therefore violating the positivity assumption.

3. The following references are highly relevant to the article and results should be discussed in relationship to these studies: Zhang, B., Fong, Y., Dang, L., Fintzi, J., Chen, S., Wang, J., ... & Coronavirus Variant Immunologic Landscape Trial (COVAIL) Study Team. (2025). Neutralizing antibody immune correlates in COVAIL trial recipients of an mRNA second COVID-19 vaccine boost. *Nature communications*, 16(1), 759.
Fong, Y., Dang, L., Zhang, B., Fintzi, J., Chen, S., Wang, J., ... & Gilbert, P. B. (2025). Neutralizing Antibody Immune Correlates for a Recombinant Protein Vaccine in the COVAIL Trial. *Clinical Infectious Diseases*, 80(1), 223-227.
Zhang, B., Fong, Y., Fintzi, J., Chu, E., Janes, H. E., Kenny, A., ... & USG/CoVPN Biostatistics Team van der Laan Lars WP 7 On behalf of the United States Government (USG) COVID-19 Immune Assays Team. (2024). Omicron COVID-19 immune correlates analysis of a third dose of mRNA-1273 in the COVE trial. *Nature Communications*, 15(1), 7954.
Sun, K., Bhiman, J. N., Tempia, S., Kleynhans, J., Madzorera, V. S., Mkhize, Q., ... & Cohen, C. (2024). SARS-CoV-2 correlates of protection from infection against variants of concern. *Nature Medicine*, 30(10), 2805-2812.

(Remarks on code availability)

Reviewer #3

(Remarks to the Author)

(Remarks on code availability)

Version 2:

Reviewer comments:

Reviewer #2

(Remarks to the Author)

Please see the uploaded Word doc.

(Remarks on code availability)

Reviewer #3

(Remarks to the Author)

I co-reviewed this manuscript with one of the reviewers who provided the listed reports. This is part of the Nature

Communications initiative to facilitate training in peer review and to provide appropriate recognition for Early Career Researchers who co-review manuscripts.

(Remarks on code availability)

REVIEWER COMMENTS

Responses to reviewers are blue.

Original manuscript text is:

Indented and black

Revised manuscript text is:

Indented and purple

Reviewer #1 (Remarks to the Author):

The study provides insights into the correlates of protection against SARS-CoV-2 Omicron variants, focusing on a population in Nicaragua which is an underrepresented region in COVID-19 research. The use of a household-based cohort study with an embedded transmission study design is particularly interesting, allowing for the tracking of virus exposure and antibody responses in a cohort that may have more consistent exposures to the virus than those in other studies. The mediation analysis is novel and attempts to assess the causal role of neutralizing antibodies in protection, rather than just as a correlate. I congratulate the authors on an interesting and well-thought-out study.

We thank the reviewer for their positive assessment.

Despite this, the general findings of the study are not particularly novel. The correlates with neutralizing antibodies are established, and it is expected that variant-specific antibodies would also be correlates, given the correlated nature of ancestral and variant-specific correlates (as shown in this article as well). Given the rapidly evolving variant landscape, the authors do not demonstrate how variant-specific correlates could be used to improve clinical practice or public health policy in practice. This limits the broader impact of this study. While the Central America focus is important to build out the understanding among all populations, the findings are mostly confirmatory, rather than creating new insight.

We agree with the reviewer that it would be helpful to emphasize the potential public health benefits of variant-specific correlates of protection. We have updated the text accordingly. Furthermore, we appreciate that the reviewer has highlighted the importance of including Central America in studies of correlates of protection. As discussed in the paper, the exposure histories of people living in Nicaragua is different than those in other countries, and therefore it was not clear at the outset of this study that the same correlates of protection would be as applicable. We have clarified this argument in the text.

For example, in the Introduction (lines 55-66):

The extent to which such evolution compromises the interpretation of anti-SARS-CoV-2 antibody assays focused on historical, non-predominant variants remains underexplored.

We hypothesize that variant-specific neutralizing antibody titers may be better correlated with protection from infections or symptoms than the standard ancestral-strain-neutralizing titers used in most assays. Also of importance to global public health, the majority of published studies of antibody correlates of protection focus on participants from North America, Europe, and Asia; relatively few include participants from Central and South America or Africa where a majority of first SARS-CoV-2 exposures were through infection, not vaccination, and less-studied vaccines were deployed. Differences in early SARS-CoV-2 exposures among individuals in Central and South America, compared to other regions raise concerns that correlates of protection identified in other populations may not be applicable in the context of Central and South America.

And in the Discussion (lines 239-243):

Our results showed that protective levels of D614G-neutralizing antibodies are likely higher than protective levels of homotypic-neutralizing antibodies, concordant with the results of Sun and colleagues, in both unvaccinated and vaccinated individuals, and supporting our hypothesis that variant-specific titers would be better correlates of protection.

The findings that are novel are those presented in the mediation analysis, where the authors aim to delve further into a causal relationship, rather than just a correlated one. However, the authors also rightfully acknowledge that this analysis carries some assumptions that limit the causal interpretations. It would be helpful if the authors elaborated on these assumptions and common ways they may be violated to aid the reader in interpreting their robustness. An example that comes to mind is whether there are other unobserved confounders between the pre-exposure titre and the outcome. This could reasonably be comorbidities, other measures of immune function, or genetic factors (given this is a household study). Any such unobserved confounders here would introduce biasing paths, which might be helpful to illustrate with further DAGs like that in Figure S3.

We thank the reviewer for their assessment of the mediation analysis. We agree that it is important to openly discuss areas where assumptions may be limiting. We addressed some of the points noted by the reviewer when designing this household-based cohort study with an embedded transmission design almost. For example, unlike most correlates of protection studies in COVID-19 literature, with titers measured months before unrecorded viral challenges, here we have titer measurements highly proximal to a known viral challenge in the household, thereby reducing the likelihood of confounding exposures. Nonetheless, the variability in comorbidities or the intriguing possibility of household-specific genetic factors could affect conclusions. We have updated the text accordingly, to specify potential confounders affecting exposures, mediators, and outcomes.

For example, in the Discussion (lines 283-290):

Sixth, potential confounders such as health-seeking behavior and risk avoidance, comorbidities, host genetic susceptibility, and socioeconomic status (SES) may influence both prior infection or vaccination (the exposure), antibody levels (the mediator), and the risk of infection (the outcome), but were not fully captured in our analysis. Finally, while our causal mediation analysis supports the role of neutralizing antibodies in mediating protection, several key assumptions of the mediation framework, including exchangeability, consistency, and positivity, may not be fully satisfied in this observational context. Therefore, causal interpretations of these results should be made cautiously.

Finally, there are some methodological concerns that need addressing for the GLM and GEM analysis. Figures S5 – S7 show that homogeneity assumptions are likely to be violated as the variance of titres is much larger for those infected than those not infected. This is fine for the univariate analysis, which is non-parametric, but there is no discussion on how this potential violation to assumptions is addressed or impacts the multivariable analysis, which are the primary analysis outcomes presented in the abstract.

We thank the reviewer for raising this important point. As shown in Figures S5–S7, we agree that the variance in pre-exposure titers differs between individuals who were infected and those who were not. However, in the context of our multivariable analyses using generalized linear models (GLMs), this unequal variance does not violate the core assumptions of logistic regression. Specifically, logistic regression does not require homoscedasticity of the predictor variables, as it models the log-odds of a binary outcome rather than the outcome itself. Additionally, we compared our logistic regression models with generalized additive models (GAMs), which allow for flexible, non-linear relationships. The similarity in results between the GLM and GAM models supports the appropriateness of the linearity assumption in the logistic regression and suggests that our specification of the relationship between pre-exposure titers and infection risk is valid.

Minor Comments:

- Numbers for protection against infection for BA.2 in the Abstract do not match the numbers in the results or Table S3. They are close, so one of these is likely a typographical error.

We thank the reviewer for pointing this out this lack of clarity. In the abstract (lines 35-38) we refer to protection from symptomatic infections (values in Table S4). However, in the Results, we previously explicitly highlighted only the results for overall infections (values in Table S3). We have updated the Results accordingly to also include explicit discussion of the values mentioned in the Abstract from Table S4.

For example, in the Results (lines 130-138):

A four-fold increase in neutralizing titers was also significantly associated with protection from symptomatic BA.1 infection: 28% protection (95% Confidence Interval [CI] 12–42%) for BA.1 neutralizing titers, 28% (CI 12–42%) for BA.2 neutralizing titers, and 28% (CI 7–

44%) for D614G neutralizing titers. For symptomatic BA.2 infection, a four-fold increase in neutralizing titers was associated with 47% protection CI 24–65%) for BA.1 neutralizing titers, 43% (CI 20–62%) for BA.2 neutralizing titers, and 57% (CI 29–75%) for D614G neutralizing titers. Although similar levels of protection were observed for each four-fold increase in neutralizing titers across assay, the average pre-exposure titer values were higher for D614G than for the Omicron sub variants (Figure S7, Table 1).

- In Figure S3, the prior vaccinations may also have a causal relationship with prior infections. I note this doesn't impact biasing paths and so is not consequential, but may be worth mentioning in the figure legend.

We thank the reviewer for this comment. We have updated the DAG and legend accordingly

For example, at the end of the legend for Figure S3 (Supplement lines 144-145):

Arrows flow from each variable to the factor or outcome it is hypothesized to influence, including between variables adjusted for, such as prior vaccinations and prior infections.

- The Nature communications form on reporting summary on statistics appears incorrect: No discussion on tests of assumptions for GLM and GEM analysis – though this is ticked as “confirmed”; The description of statistical parameters is not complete. Means and CIs are presented, but not standard deviations.; P-values are presented, but not their associated test statistics and degrees of freedom (for GLM/GEM analysis).

We tested the assumption of independence by adjusting for household IDs, and the linearity of the odds ratios by comparing GAM and GLM outputs. We have updated the Methods section accordingly.

For example, in Methods (lines 392-401):

We also assessed the multivariate association between titers and infection outcomes by Omicron wave and assay with generalized linear models (GLMs) under a binomial family. We tested the assumption of independence by adjusting for household IDs, and the linearity of the odds ratios by comparing GAM and GLM outputs. Two sensitivity analyses were added to assess the robustness of the multivariable findings. First, we applied a Generalized Estimating Equations (GEE) framework to account for within-household clustering, using household ID as the clustering variable. Second, we extended the multivariable model to additionally adjust for time since last exposure (infection or vaccination), which may influence both antibody levels and risk of reinfection.

Regarding descriptions of statistical parameters, we originally presented means, CIs, and p-values. We thank the reviewer for pointing out a place with omitted values. We have added degrees of freedom, standard errors, and Z statistics to appropriate tables. Please see Tables S3-S9 for updates.

Reviewer #1 (Remarks on code availability):

Code was unavailable to be reviewed as it was not part of the reviewing pack, and does not appear to be publicly available.

We have published the code in a publicly accessible repository with a DOI via Zenodo, and we have updated the Statistical Analysis section of the Methods accordingly (lines 427-428). Please see <https://doi.org/10.5281/zenodo.15749005>.

Reviewer #2 (Remarks to the Author):

The dataset reported this manuscript is unique and provides a valuable opportunity to study the relationship between SARS-CoV2-specific humoral immune responses and risk of COVID.

We thank the reviewer for their positive assessment.

Major comments:

1. The statistical results for the two endpoints, symptomatic and asymptomatic infections, are qualitatively the same, as can be seen from comparing panel A and B in Figure 1. It could make a clearer presentation if the main text focuses on one set of endpoints and the supplementary results cover the other.

We thank the reviewer for this suggestion. We agree that the trends are similar across the two endpoints, and that the figures may appear crowded as a result. However, for several reasons, we think it remains critical to keep both sets of results in the main figures. First, a correlate of protection against overall infection has different public health implications than a correlate of

protection against symptomatic infection alone. For example, protection from overall infection, indicative of sterilizing immunity, in models of infection outbreaks, would imply different levels of susceptibility and eventual ongoing transmission frequency than protection from only symptomatic infection. Meanwhile, protection from symptomatic infection is the correlate that is most relevant to counseling of individual patients concerned about developing upper respiratory infections. Second, there are meaningful statistical differences between our results for overall infections and symptomatic infections. In the BA.1 wave, we did not observe a significant protective neutralizing titer threshold for overall infections, while there were clear, significant protective thresholds observed for symptomatic infection. However, in the BA.2 wave, we observed significant protective thresholds for both overall and symptomatic infections. Nonetheless, while we think it is important to highlight both conclusions in the main text given the interest in correlates of protection from both any infection and symptomatic infection, we fully agree with the reviewer that it is important to have clear and interpretable results that are not over-crowded. Therefore, we have updated the figures to more clearly delineate which plots address any infection or symptomatic infection.

The updated Figure 1 is:

2. Authors defined 'protection' by $1 - OR$, where OR is the coefficient of the titer level in a GLM with a binary outcome (infection or not). The language is confusing. It is perhaps clearer to report it as association.

We appreciate the reviewer's comment and understand that expressing protection as $1 - OR$ may be unfamiliar to some readers. However, this approach is widely used and accepted in the vaccine and correlates of protection literature to quantify the relative reduction in odds of infection per unit increase in the antibody marker. This formulation provides a more intuitive understanding of the protective association of immune responses. This is how protection conferred by vaccines, i.e., vaccine effectiveness, is calculated. Therefore, it is more directly interpretable as a correlate of protection than an association.

That said, we recognize the potential for confusion among audiences less familiar with this convention. To enhance clarity, we have clarified this definition in the Methods section and ensured that the plain odds ratios (ORs) and their corresponding confidence intervals are also provided in the supplementary tables for transparency and accessibility.

For example, in the Methods (lines 403-405):

Protection associated with increasing titer levels was quantified as $1 - \text{odds ratio (OR)}$ to quantify the relative reduction in odds of infection per unit increase in the antibody marker, holding covariates fixed, similar to calculations of vaccine effectiveness.

Lastly, we agree that the way we phrased descriptions of some statistical analyses throughout was a bit confusing, in some places. Therefore, we have updated some wording to ensure that we precisely described protective thresholds of neutralizing antibody titers and the effect sizes of increases in antibody titers.

For example, in the Results (lines 130-135):

A four-fold increase in neutralizing titers was also significantly associated with protection from symptomatic BA.1 infection: 28% protection (95% Confidence Interval [CI] 12–42%) for BA.1 neutralizing titers, 28% (CI 12–42%) for BA.2 neutralizing titers, and 28% (CI 7–44%) for D614G neutralizing titers. For symptomatic BA.2 infection, a four-fold increase in neutralizing titers was associated with 47% protection (CI 24–65%) for BA.1 neutralizing titers, 43% (CI 20–62%) for BA.2 neutralizing titers, and 57% (CI 29–75%) for D614G neutralizing titers.

3. On the first objective, the main results are that the antibodies are not correlates in the BA.1 wave but are correlates in the BA.2 wave. This is an interesting and intriguing pair of results. We find the description of these two results unclear and the discussion of the results lacking. Is it true that we can think of it as two separate datasets close in time? Would the two datasets be from the same population, i.e. wouldn't the BA.1 wave change the immunity of the population? For each dataset, should the household be treated as a cluster in regression? How do we

interpret the coefficients for antibody markers when infection history and vaccine history, which affect immune biomarker levels, are adjusted in regression?

We thank the reviewer for their enthusiasm for these results and for their suggestions. We agree with the reviewer that the difference in results for the BA.1 and BA.2 waves could at least partially be explained by more potential Omicron exposures prior to BA.2 household invasions than prior to BA.1.

In response to the reviewer's suggestions, we have conducted two additional sensitivity analyses. First, we applied a Generalized Estimating Equations (GEE) framework with household ID included as a clustering variable to account for within-household correlation. Second, we further adjusted the regression model for time since last exposure (either vaccination or infection), in addition to household ID. In both sensitivity analyses, the results were consistent with our primary findings, supporting the robustness of the observed associations. These additional analyses and their results are now described in the revised Methods and Results sections and included in new Supplementary Tables S5-8.

For example, in the Methods (lines 391-401):

Univariate analyses using Wilcoxon rank sum test were applied to examine the relationship between antibody titer and infection outcome by assay and Omicron wave (BA.1 or BA.2). We also assessed the multivariate association between titers and infection outcomes by Omicron wave and assay with generalized linear models (GLMs) under a binomial family. Two sensitivity analyses were added to assess the robustness of the multivariable findings. First, we applied a Generalized Estimating Equations (GEE) framework to account for within-household clustering, using household ID as the clustering variable. Second, we extended the multivariable model to additionally adjust for time since last exposure (infection or vaccination), which may influence both antibody levels and risk of reinfection.

In the Results (lines 135-139):

Although similar levels of protection were observed for each four-fold increase in neutralizing titers across assay, the average pre-exposure titer values were higher for D614G than for the Omicron sub variants (Figure S7, Table 1). Results did not differ after adjusting for household clustering in this household design and further adjusting for time since last exposure (Tables S5-8).

We have also revised the Discussion to clarify the temporal relationship between the BA.1 and BA.2 waves, and the implications for interpreting correlates of protection in a dynamically changing immune landscape.

For example, in the Discussion (lines 277-281):

Third, our analysis to detect correlates of protection for BA.1 infection may be underpowered. Fourth, there may be a difference in the number of unreported SARS-CoV-2 exposures for participants in the earlier BA.1 and later BA.2 waves, which could affect the difference in correlation with protection during these two waves.

Lastly, we refer to the response to point 2 to clarify interpretation of antibody titer coefficients after adjustment for other exposures (lines 403-405):

Protection associated with increasing titer levels was quantified as $1 - \text{odds ratio (OR)}$ to quantify the relative reduction in odds of infection per unit increase in the antibody marker, holding covariates fixed, similar to calculations of vaccine effectiveness.

4. On the second objective, the mediation analysis, the presentation is very unclear, which makes the results difficult to interpret. A reference example may be the mediation section in <https://www.science.org/doi/10.1126/science.abm3425>. Some specific comments: First, is prior infection and prior vaccination treated separately or together? Second, in the cohort, according to Table 1, there are only 23 (6.7%) ppts with no prior infection. How many got infected among these 23 ppts? If the authors thought of 'prior infection' as a treatment, the treatment is very imbalanced. Third, time since last infection is heterogeneous and this implies the 'direct effect' from prior infection on infection is heterogeneous. Not sure if authors considered this in the mediation analysis. Fourth, mediation analysis relies on a 'positivity assumption' of the mediator. Again, this is likely not satisfied. Put together, I do not find the proportion mediated reported by the authors convincing.

We thank the reviewer for their detailed and thoughtful comments on the mediation analysis. Prior vaccination and prior infection are analyzed as the exposure separately, with adjustment for the other in each respective analysis. For example, if prior vaccination is the exposure, then it was adjusted for prior infection. We have added additional Methods details for clarity. We thank the reviewer for pointing us to the concise presentation in Table S9 of the Supplement of Gilbert et al., 2022. In order to convey the most statistics that readers may find informative of mediated effects, we included both RR and Protection in our tables, with respective confidence intervals, for direct, indirect, and total effects.

In the Methods (lines 417-420):

For these analyses, we combined both Omicron waves and adjusted for Omicron wave, age and prior infection or vaccination (as main exposure or a confounder, as appropriate). That is, when prior infection was the exposure, vaccination was considered a confounder, and vice versa.

We agree that the assumptions underlying mediation analysis, particularly positivity, are challenging to fully satisfy in observational studies, especially in high-transmission settings such as Nicaragua's given the study period. As noted, only 6.7% of participants in our cohort had no

evidence of prior infection, and the positivity assumption is likely violated. We now explicitly acknowledge this limitation in the revised Discussion.

For example (lines 286-289):

Finally, while our causal mediation analysis supports the role of neutralizing antibodies in mediating protection, several key assumptions of the mediation framework, including exchangeability, consistency, and positivity, may not be fully satisfied in this observational context.

Nonetheless, we believe that the question of whether antibody markers mediate protection conferred by prior infection is important to examine, especially in real-world populations with high force of infection (F₀) such as in Nicaragua. Despite the imbalance in prior infection status, the results followed the expected direction of association and were robust to non-parametric bootstrapping, which we used to address statistical uncertainty given the small sample size in some strata.

We also recognize the reviewer's point regarding heterogeneity in time since last infection. While we did not stratify by time since infection in the mediation analysis, we have now included a sensitivity analysis adjusting for time since last exposure in the outcome model (Tables S7,8).

Minor comments:

1. The literature research lacks references to some more recent (2023-present) papers on Omicron correlates.

We thank the reviewer for bringing this to our attention. We have completed additional literature review and added references to the discussion accordingly (line 190). These additional references include Marking et al., 2023; Zhong et al., 2024, and Canetti et al., 2024. Marking et al., 2023 showed that fully-neutralizing titers against live WT virus were not informative of protection from Omicron infections in their cohort of vaccinated healthcare workers in Sweden. Zhong et al., 2024 showed that WT- or BA.2-neutralizing antibody titers were correlated with protection from symptomatic Omicron infection in a small cohort of children with hybrid immunity in Singapore. Canetti et al., 2024 showed that XBB variant-specific nAb titers were more informative as a correlate of protection than WT-neutralizing titers in a cohort of healthcare workers in Israel.

2. Authors said they determined the 'threshold' (see methods) using splines. The supplementary tables define what threshold means. That detail should be moved to the methods.

Thank you. We have now moved the definition of threshold into the methods section. For example (lines 405-408):

We also determined protective cut-offs or thresholds to achieve 50 and 80% protection using GLM and Generalized Additive Models (GAM) with thin-plate spline regression with three knots to allow for non-linear effects.

Reviewer #3 (Remarks to the Author):

We thank the reviewer for their comments.

Response to Reviewers

Reviewer #1 (Remarks to the Author):

The authors have satisfactorily addressed my comments from the prior review and I have no further comments, apart from comments on code availability.

We thank the reviewer for their positive assessment.

Reviewer #1 (Remarks on code availability):

The submitted code comprises approximately 1,500 lines. While a full line-by-line review was not feasible, a general assessment indicates that the code is well-organized, clearly formatted, and adequately commented. These attributes suggest that the code should be understandable to readers with a working knowledge of R. Spot checks did not reveal any coding errors or issues.

We thank the reviewer for their positive assessment.

The availability of the code significantly supports the reproducibility of the methods described in the manuscript. To further enhance reproducibility, it is recommended that the authors provide dummy datasets compatible with the code. This would allow reviewers and readers to compile and validate the code independently. For full reproducibility, access to the original datasets used in the study would be necessary. However this depends on whether the data can be made publicly available.

We appreciate the reviewer's suggestion. We have added a dummy data set of synthetic data to the repository to demonstrate the usability of the code. Code and the synthetic dataset can be found here: <https://zenodo.org/records/16804602>. Additionally, as described in the Data Availability section, since the data include information collected in Nicaragua, access is subject to Nicaraguan data ownership regulations and may require approval from relevant Nicaraguan authorities. Therefore, the full dataset needed for complete reproduction of the figures is available upon requests that are approved by the relevant Nicaraguan authorities, Aubree Gordon (corresponding author) and the Health Sciences and Behavioral Sciences Institutional Review Board at the University of Michigan. This is consistent with ethical and legal frameworks governing data use in international collaborative studies.

Reviewer #2 (Remarks to the Author):

We appreciate the authors' detailed responses. Some remaining concerns are listed below.

1. The authors' response did not cite any references to support their claim that the use of the term "protection" for 1-OR is widely accepted in today's literature. I looked up the three new references the authors added. None used protection to refer to 1-OR: Marking et al. reported

RR, Zhong et al. reported area under ROC, Canetti et al. reported OR. Moreover, the x axis and the y axis labels in Fig 1 do not work together. For example, the first panel seems to indicate that the protection is around 25% when the BA.1 neut titer is 1, which does not make much sense. A similar plot in Canette et al. can be used as a example of a truer representation of the results. Canette et al. Fig 2B labels the y axis as OR and the caption reads: "Estimated odds ratio of infection as a function of antibody levels compared to an individual with undetectable antibody levels, based on a crude logistic regression model." The key here is that the OR is a contrast and it is associated with a contrast in titer, not in absolute titer.

We appreciate the reviewer's careful consideration of the results. The connection between the term "protection" and the statistic "1 - odds ratio" is established in the literature for a variety of exposures and interventions. However, we of course agree that the three papers we cited as additional recent examples of SARS-CoV-2 correlates of protection studies did not happen to use this statistic. We note that the term "protection" is commonly used when explaining vaccine effectiveness, and that vaccine effectiveness is often quantified as 1-OR, comparing the vaccinated population relative to the unvaccinated population in observational studies ¹⁻⁵. Additionally, 1-OR has been used to convey the protective effects conferred by non-vaccination exposures, such as unrelated prior infections ⁶ or insect repellent ⁷. Further, 1-OR has been used to communicate levels of protection in studies of other immune correlates of protection against other respiratory infections ⁸. To facilitate interpretability and support comparison, we also report odds ratio (OR) estimates in all analyses (Tables S3-8 and Table S11).

On Figure 1, to improve clarity, we have revised the y-axis label. While the original label read "Protection (%)", more precisely, the figure displays the predicted probability of remaining uninfected, estimated as (1 - predicted probability of infection) from a fitted GAM logistic regression model. These probabilities reflect risk as a function of antibody titers, based on a fixed covariate profile (vaccinated adults with a prior infection), and are not odds ratios or contrasts relative to an "undetectable titer" or "lower titer" group. There is no contradiction, therefore, in having the GAM output shown over the range of titers displayed. We have updated the figure legend and axis labels accordingly to reflect this interpretation better and avoid confusion. Figure legends now read "Predicted % not infected" and "Predicted % not symptomatic" accordingly.

We also appreciate the comparison to Canetti et al., where odds ratios (ORs) are plotted relative to an undetectable baseline. However, in our setting, we opted not to use "undetectable" titers as a reference group because most participants had prior SARS-CoV-2 exposure, few participants have undetectable titers, and an undetectable titer does not reliably indicate participants are truly immunologically naive. Accordingly, we report 1 - ORs, interpreted as the reduction in risk per four-fold increase in antibody titer, in the Supplement (Figure S7, Tables S3-S8, S11), and in Figure 1 we model the reduction in risk and show the corresponding 50% and 80% protection thresholds (Tables S9-10) as a function of titer on a linear scale (with \log_4 -transformed titers so that one unit equals a four-fold change).

We have changed the language in several parts of the manuscript and supplement to clarify this.

In the Results (lines 126-127):

Similarly, after adjusting for age, infection history and vaccine history, we found that a four-fold linear increase in neutralizing titers across neutralization assays was significantly associated with protection from BA.2 infection—corresponding to 45% protection (95% Confidence Interval [CI] 21–64%) for BA.1 neutralizing titers, 45% (CI 21–64%) for BA.2 neutralizing titers, and 51% (CI 23–71%) for D614G neutralizing titers—but not with protection from BA.1 infection (Figure S7, Table S3). Here, protection is interpreted as the relative reduction in the odds of infection per four-fold increase in antibody titers, calculated as $1 - \text{OR} \times 100$.

In the methods section (lines 392-400)

Protection associated with increasing titer levels was quantified as $1 - \text{odds ratio (OR)} \times 100$ to quantify the relative reduction in odds of infection or symptomatic infection per a 4-fold increase in the antibody marker, holding covariates fixed, similar to calculations of vaccine effectiveness. We also determined protective cut-offs or thresholds to achieve 50 and 80% protection (reduction in risk) using multivariate GLM and Generalized Additive Models (GAM) with thin-plate spline regression with three knots to allow for non-linear effects. Multivariate GAM and GLM models were adjusted for the confounders mentioned above. Protective thresholds were estimated by holding covariates constant, specifically representing vaccinated adults with prior natural infection, who, for the threshold analysis, constituted our primary population of inference.

In Fig. 1 legend (lines 567-576):

Figure 1. Neutralizing titers as a correlate of protection by Omicron wave and assay. Relationship between neutralizing titers (ID50) and protection against BA.1 and BA.2 infection (A) and symptomatic infection (B) during their respective waves. The solid lines represent the predicted protection curves, calculated as $1 - \text{predicted risk}$, according to a Generalized Additive Model (see Methods), with the shaded ribbon bands denoting the 95% confidence intervals (CIs) for these estimates, derived from vaccinated and prior-infected adults. The horizontal blue solid lines mark the points where the predicted protection curves cross the 50% and 80% protection thresholds, highlighting the titer levels required to achieve these levels of protection. Embedded within the figure are histograms showing the raw distribution of neutralization titers, stratified by outcome.

We also added a note on the footnotes for Tables S9 and S10 to clarify how thresholds were estimated:

TS: Threshold. Protective thresholds to achieve 50% or 80% reduction in risk were derived from multivariate model estimate predictions based on vaccinated adults with prior natural infections.

2. Violation of the positivity assumption in a causal mediation analysis typically means the probability of the mediator taking any level conditional on the baseline covariates bounded away from 0; see, e.g., Identification, Inference and Sensitivity Analysis for Causal Mediation Effects. For instance, depending on baseline covariates like previous vaccine and infection history, the mediator level may be poorly overlapped and therefore violating the positivity assumption.

We agree with the reviewer and have enhanced our comment regarding the positivity assumption violation in the Limitations subsection of the Discussion accordingly.

In the Discussion (lines 276-283):

Finally, while our causal mediation analysis supports the role of neutralizing antibodies in mediating protection, several key assumptions of the mediation framework, including exchangeability, consistency, and positivity, may not be fully satisfied in this observational context. In particular, the positivity assumption may be violated due to the very low prevalence of individuals without prior SARS-CoV-2 infection in this cohort. This lack of adequate representation of uninfected participants reduces overlap between exposure groups, which can result in unstable or biased mediation effect estimates. Therefore, causal interpretations of these results should be made cautiously.

3. The following references are highly relevant to the article and results should be discussed in relationship to these studies:

Zhang, B., Fong, Y., Dang, L., Fintzi, J., Chen, S., Wang, J., ... & Coronavirus Variant Immunologic Landscape Trial (COVAIL) Study Team. (2025). Neutralizing antibody immune correlates in COVAIL trial recipients of an mRNA second COVID-19 vaccine boost. *Nature communications*, 16(1), 759.

Fong, Y., Dang, L., Zhang, B., Fintzi, J., Chen, S., Wang, J., ... & Gilbert, P. B. (2025). Neutralizing Antibody Immune Correlates for a Recombinant Protein Vaccine in the COVAIL Trial. *Clinical Infectious Diseases*, 80(1), 223-227.

Zhang, B., Fong, Y., Fintzi, J., Chu, E., Janes, H. E., Kenny, A., ... & USG/CoVPN Biostatistics Team van der Laan Lars WP 7 On behalf of the United States Government (USG) COVID-19 Immune Assays Team. (2024). Omicron COVID-19 immune correlates analysis of a third dose of mRNA-1273 in the COVE trial. *Nature Communications*, 15(1), 7954.

Sun, K., Bhiman, J. N., Tempia, S., Kleynhans, J., Madzorera, V. S., Mkhize, Q., ... & Cohen, C. (2024). SARS-CoV-2 correlates of protection from infection against variants of concern. *Nature Medicine*, 30(10), 2805-2812.

We thank the reviewer for pointing us to the additional references for discussion, and we apologize for the omission of the relevant COVE and COVAIL citations. We have added these to the paper, in addition to our existing discussion of Sun and colleagues' paper, in the

Discussion. The COVAIL papers and Sun et al. cited above include neutralization assays targeting multiple strains, while the COVE paper only includes new neutralization data for BA.1.

In the Discussion (lines 186-187):

Many prior studies have demonstrated, to varying degrees, that humoral immunity serves as a correlate of protection from SARS-CoV-2 infection or severe symptoms.^{2-7,9,15,21,23-28}

And Discussion (lines 218-224):

Relatively few correlates of protection studies have been published with such concurrent analyses of neutralizing antibodies directed against different prior and contemporaneous variants, and none have been household-based cohort with an embedded transmission study triggered by defined index case exposures around the time of titer measurement like this study.^{15,26,27,32,33} Recent COVAIL trial reports similarly identified neutralizing antibodies as correlates of protection against Omicron across mRNA and recombinant protein vaccines. Our household-based study extends these findings to a community setting where heterogeneous exposures, including multiple infections and vaccines less frequently evaluated globally.^{26,27}

Reviewer #3 (Remarks to the Author):

Reviewer #3 (Remarks on code availability):

Document References

1. Shapiro, E. D. & Clemens, J. D. A controlled evaluation of the protective efficacy of pneumococcal vaccine for patients at high risk of serious pneumococcal infections. *Ann. Intern. Med.* **101**, 325–330 (1984).
2. Butt, A. A., Omer, S. B., Yan, P., Shaikh, O. S. & Mayr, F. B. SARS-CoV-2 vaccine

effectiveness in a high-risk national population in a real-world setting. *Ann. Intern. Med.* **174**, 1404–1408 (2021).

3. Quinn, H. E., Snelling, T. L., Macartney, K. K. & McIntyre, P. B. Duration of protection after first dose of acellular pertussis vaccine in infants. *Pediatrics* **133**, e513–9 (2014).
4. Huh, K. *et al.* Risk of severe COVID-19 and protective effectiveness of vaccination among solid organ transplant recipients. *J. Infect. Dis.* **229**, 1026–1034 (2024).
5. Sullivan, S. G. & Kelly, H. Stratified estimates of influenza vaccine effectiveness by prior vaccination: caution required. *Clin. Infect. Dis.* **57**, 474–476 (2013).
6. Nacher, M. *et al.* Helminth infections are associated with protection from cerebral malaria and increased nitrogen derivatives concentrations in Thailand. *Am. J. Trop. Med. Hyg.* **66**, 304–309 (2002).
7. Rowland, M. *et al.* DEET mosquito repellent provides personal protection against malaria: a household randomized trial in an Afghan refugee camp in Pakistan. *Trop. Med. Int. Health* **9**, 335–342 (2004).
8. Tsang, T. K. *et al.* Investigation of CD4 and CD8 T cell-mediated protection against influenza A virus in a cohort study. *BMC Med.* **20**, 230 (2022).

We are encouraged by the thoughtful responses from the authors.

To provide some further clarification to our comment about the use of the term “protection”, we fully agree with the authors that there is a “connection between the term protection and the statistic 1 - odds ratio.” What we are saying is that that is an interpretation of the OR. It is rare to label 1-OR as protection in axis labels in figures or column headers in tables. For example, this distinction can be seen in the following table from Huh, K. et al. (J. Infect. Dis. 2024), one of the new references cited by the authors. Note that the caption uses the term effectiveness, but the header says OR:

[REDACTED]

We appreciate the reviewer’s perspective and the example from Huh et al. We agree that the underlying statistic we are reporting is the odds ratio, and we recognize that one conventional practice is to present ORs explicitly in table headers and figure axes. At the same time, our decision to use the label “Protection (1 - OR)” was deliberate. In studies of immune correlates, the concept of protection, operationalized as the reduction in odds of infection or disease associated with higher antibody levels—is central to interpretation. We considered that presenting protection explicitly in labels would make the protective interpretation of the estimates more transparent to readers, particularly those from the immunology and vaccine fields who are accustomed to thinking in terms of protection rather than odds ratios.

The authors opted to change the dependent variable in the plot to risk. We like that change. However, we wonder whether the sampling design supports the estimation of risk. It is well known that logistic regression gives OR estimates but not risk estimates for some sampling designs. From the following paragraph in the paper, it is not clear whether the 345 individuals represented a sample from the cohort or the cohort itself. Could the authors clarify?

93 We measured SARS-CoV-2 antibody levels in 345 individuals from the HICS study before
94 exposure to BA.1 or BA.2 and used these measurements to assess the protective effect of pre-
95 existing antibodies against infection and moderate/severe infection. Households included in the
96 study experienced the introduction of BA.1 or BA.2 SARS-CoV-2 strains by a household

We thank the reviewer for raising this point. On the issue of logistic regression and risk estimation, we note that our household study includes all exposed contacts of each index case rather than a case–control sampling design. Therefore, it is appropriate to estimate risk directly. The 345 individuals described represent the full set of participants enrolled in the household cohort who were exposed to BA.1 or BA.2 introductions during the study period, not a subsample. Thus, our sampling design supports risk estimation.

Current text:

“We measured SARS-CoV-2 antibody levels in 345 individuals from the HICS study before exposure to BA.1 or BA.2 and used these measurements to assess the protective effect of pre-existing antibodies against infection and moderate/severe infection.”

Revised text (lines 93-96):

“We measured SARS-CoV-2 antibody levels in 345 individuals from the HICS study, representing all enrolled household contacts of each index case exposed to BA.1 or BA.2 introductions, before exposure to these variants. We used these measurements to assess the protective effect of pre-existing antibodies against infection and moderate/severe infection.”

Finally, we are sorry to bring this up at this late stage, but it is a somewhat serious technical issue, and we missed it by chance before. The highlighted sentence in the paragraph below suggested that household IDs were adjusted. It reminds us of the Neyman-Scott problem, where the number of model parameters increases at the same order with the sample size. Could the authors clarify?

379 *Statistical analysis*

380 Univariate analyses using Wilcoxon rank sum test were applied to examine the relationship
381 between antibody titer and infection outcome by assay and Omicron wave (BA.1 or BA.2). We
382 also assessed the multivariate association between titers and infection outcomes by Omicron
383 wave and assay with generalized linear models (GLMs) under a binomial family. We tested the
384 assumption of independence by adjusting for household IDs, and the linearity of the odds ratios
385 by comparing GAM and GLM outputs. Two sensitivity analyses were added to assess the
386 robustness of the multivariable findings. First, we applied a Generalized Estimating Equations
387 (GEE) framework to account for within-household clustering, using household ID as the
388 clustering variable. Second, we extended the multivariable model to additionally adjust for time
389 since last exposure (infection or vaccination), which may influence both antibody levels and risk
390 of reinfection.
391

9

We appreciate the reviewer’s careful attention to this detail. It is understandable that this point may have been missed previously, as the sentence was added later in response to Reviewer 1’s recommendation, and as a supplementary analysis (Tables S5-8 only). With respect to household clustering, we adjusted for household IDs and conducted

sensitivity analyses using robust standard errors and generalized estimating equations (GEE) frameworks. These approaches are widely applied in household-based infectious disease studies and help mitigate concerns of the Neyman–Scott type. Specifically, they provide stable estimates without the parameter proliferation that would arise from fitting household fixed effects.

To prevent any misinterpretation, we have clarified this in the Methods section. In addition, results show overall low correlation parameters, suggesting that household-level effects were relatively low compared to individual-level variation (Tables S5-S8). We have added text in the Results section to reflect this finding.

Current Text on methods:

First, we applied a Generalized Estimating Equations (GEE) framework to account for within-household clustering using household ID as the cluster variable. Second, we extended the multivariable model to additionally adjust for time since last exposure (infection or vaccination), which may influence both antibody levels and risk of reinfection.

Revised text on methods (lines 389-394):

First, we applied a Generalized Estimating Equations (GEE) framework with an exchangeable working correlation structure, clustering on household ID. Robust (sandwich) standard errors and 95% confidence intervals were extracted to account for within-household correlation. Second, we extended the multivariable model to additionally adjust for time since last exposure (infection or vaccination), which may influence both antibody levels and risk of reinfection.

Current text on results:

Results did not differ after adjusting for household clustering in this household design and time since last exposure (Tables S5-S8).

Revised text on results (lines 141-144):

Results did not differ after adjusting for household clustering in this household design and time since last exposure. The estimated within-household correlation parameters were low, suggesting that variation in infection risk was driven primarily by individual-level rather than household-level effects (Tables S5-8).

We are encouraged by the thoughtful responses from the authors.

To provide some further clarification to our comment about the use of the term “protection”, we fully agree with the authors that there is a “connection between the term protection and the statistic 1 - odds ratio.” What we are saying is that that is an interpretation of the OR. It is rare to label 1-OR as protection in axis labels in figures or column headers in tables. For example, this distinction can be seen in the following table from Huh, K. et al. (J. Infect. Dis. 2024), one of the new references cited by the authors. Note that the caption uses the term effectiveness, but the header says OR:

[REDACTED]

The authors opted to change the dependent variable in the plot to risk. We like that change. However, we wonder whether the sampling design supports the estimation of risk. It is well known that logistic regression gives OR estimates but not risk estimates for some sampling designs. From the following paragraph in the paper, it is not clear whether the 345 individuals represented a sample from the cohort or the cohort itself. Could the authors clarify?

93 We measured SARS-CoV-2 antibody levels in 345 individuals from the HICS study before
94 exposure to BA.1 or BA.2 and used these measurements to assess the protective effect of pre-
95 existing antibodies against infection and moderate/severe infection. Households included in the
96 study experienced the introduction of BA.1 or BA.2 SARS-CoV-2 strains by a household

3

97 member, with these strains circulating consecutively between January and June 2022 (Figure
98 S2). The final sample consisted of 194 individuals who were later infected by either the BA.1 (N
99 = 153) or BA.2 (N = 41) strains, and 151 individuals who remained uninfected. Study

Finally, we are sorry to bring this up at this late stage, but it is a somewhat serious technical issue, and we missed it by chance before. The highlighted sentence in the paragraph below suggested that household IDs were adjusted. It reminds us of the Neyman-Scott problem, where the number of model parameters increases at the same order with the sample size. Could the authors clarify?

379 *Statistical analysis*

380 Univariate analyses using Wilcoxon rank sum test were applied to examine the relationship
381 between antibody titer and infection outcome by assay and Omicron wave (BA.1 or BA.2). We
382 also assessed the multivariate association between titers and infection outcomes by Omicron
383 wave and assay with generalized linear models (GLMs) under a binomial family. We tested the
384 assumption of independence by adjusting for household IDs, and the linearity of the odds ratios
385 by comparing GAM and GLM outputs. Two sensitivity analyses were added to assess the
386 robustness of the multivariable findings. First, we applied a Generalized Estimating Equations
387 (GEE) framework to account for within-household clustering, using household ID as the
388 clustering variable. Second, we extended the multivariable model to additionally adjust for time
389 since last exposure (infection or vaccination), which may influence both antibody levels and risk
390 of reinfection.
391

9